# The Electroencephalogram (EEG) Study for Estimating Endurance Sports Performance Base on Eigenvalues Extraction Method

**DOI:** 10.3390/brainsci14111135

**Published:** 2024-11-12

**Authors:** Zijian Zhou, Hongqi Xu, Yubing Sun, Guangda Liu

**Affiliations:** 1Research Field of Medical Instruments and Bioinformation Processing, College of Instrumentation and Electrical Engineering, Jilin University, No. 938 West Democracy Street, Changchun 130061, China; zjzhou21@mails.jlu.edu.cn (Z.Z.); sunyb22@mails.jlu.edu.cn (Y.S.); 2Research Center of Exercise Capacity Assessment and Promotion, School of Sports Science and Physical Education, Northeast Normal University, Changchun 130024, China; xuhq375@nenu.edu.cn

**Keywords:** cycling, electroencephalography, elite athletes, individual alpha peak frequency, shannon entropy

## Abstract

Objectives. Brain–behavior connections are a new means to evaluate sports performance. This electroencephalogram (EEG) study aims to estimate endurance exercise performance by investigating eigenvalue trends and comparing their sensitivity and linearity. Methods. Twenty-three cross-country skiers completed endurance cycling tasks. Twenty-four-channel full-brain EEG signals were recorded in the motor phase and recovery phase continuously. Eighteen EEG eigenvalues calculation methods were collected, commonly used in previous research. Time-frequency, band power, and nonlinear analyses were used to calculate the EEG eigenvalues. Their regression coefficients and correlation coefficients were calculated and compared, with the linear regression method. Results. The time-frequency eigenvalues shift slightly throughout the test. The power eigenvalues changed significantly before and after motor and recovery, but the linearity was not satisfactory. The sensitivity and linearity of the nonlinear eigenvalues were stronger than the other eigenvalues. Of all eigenvalues, Shannon entropy showed completely non-overlapping distribution intervals in the regression coefficients of the two phases, which were −0.1474 ± 0.0806 s^−1^ in the motor phase and 0.2560 ± 0.1365 s^−1^ in the recovery phase. Shannon entropy amplitude decreased more in the F region of the brain than in the other regions. Additionally, the higher the level of sport, the slower the decline in Shannon entropy of the athlete. Conclusions. The Shannon entropy method provided more accurate estimations for endurance exercise performance compared to other eigenvalues.

## 1. Introduction

Psychological factors play a crucial role in athletic performance as they are closely linked to an athlete’s response, selection, and inhibition [1,2,3]. In endurance exercise, an athlete’s mental activity has a continuous impact on performance [4]. Endurance exercises are generally defined as those which require the maintenance of a high cardiac output over extended periods of time [5]. Endurance sports are competitive endurance exercises [6]. Therefore, it is essential to have a rapid and continuous method for monitoring an athlete’s psychological state to plan training and competition strategies for endurance athletes. Historically, many psychological monitoring methods, such as action observation, motor imagery [7], and mental fatigue monitoring [8], have been developed based on Electroencephalography (EEG) [9]. EEG is a powerful means for studying brain–behavior connections and has been widely used in the study of attention, cognition, and the detection of neurological diseases [10,11]. In particular, EEG measurements adapt to outdoor testing requirements [10], have an elevated temporal resolution [7], and avoid ethical concerns [11], supporting EEG as the most commonly used technique to assess brain activity and sports psychology in the face of the challenges of serving competitive sports.

Endurance exercise is unique in its psychological demands compared to other forms of exercise. Athletes must divert their attention away from their internal sensations to improve their economy of motion [12]. This differs from findings in the motion control field, where neglecting one’s posture is known to result in poor athletic performance [13]. Several studies have shown that the EEG pattern of subjects during endurance exercise differs from that during other exercises [14,15]. Specifically, an acute bout of exhaustive endurance exercise can shift subjects’ individual alpha peak frequency (IAPF) of EEG to a higher frequency compared to non-endurance exercise [14]. Furthermore, long-term endurance exercise has been shown to have positive effects on cognition [16,17]. Therefore, it remains to be tested whether EEG rules generated by non-endurance exercise can be applied to endurance exercise. Currently, EEG studies that focus on endurance exercise and aim to enhance the abilities of endurance athletes are limited due to neglecting the differences between endurance exercise and other types of exercise. Endurance sports are a crucial component of competitive sports, and athletes’ endurance capacity is one of the fundamental criteria for evaluation [18]. Additionally, given the continuous impact of athletes’ mental activity on athletic performance, quantifying and monitoring mental activity is highly valuable.

Eigenvalue extraction is a frequently used method with high temporal resolution that is suitable for monitoring the effects of prolonged events on the EEG [19,20]. EEG eigenvalues can be divided into three categories: time-frequency eigenvalues, power eigenvalues, and nonlinear eigenvalues, according to the calculation method. Time-frequency eigenvalues, were among the earliest methods used in EEG research. Power eigenvalues refer to the power of EEG in specific frequency bands or the relative power between different frequency bands, such as delta, theta, alpha, and other rhythms. This method is commonly used in EEG research because the four main EEG rhythms have been shown to correlate with cortical activities [8,11]. Nonlinear eigenvalue methods are related to stochastic mechanisms, which are two different directions in time series analysis from determinism [21]. The output variables of physiological systems exhibit fluctuations that contain not only noise contamination and relevant event-stimulated signals, but also indirect reflections caused by complex regulatory mechanisms [22,23]. Therefore, investigating stochastic mechanisms contributes to studying brain–behavior connections from a holistic perspective. Due to the brain’s complex structure and operating mechanisms [21], most events have a weak effect on it, which leads to the fact that stochastic mechanism studies usually yield significantly better results than deterministic studies.

The purpose of the present paper is to investigate differences in EEG eigenvalue trends in athletes of different skill levels during endurance exercise. EEG data were collected from twenty-three cross-country skiers during endurance cycling tasks, and eighteen eigenvalues were calculated. In accordance with the economy of motion strategy during endurance sports [2], athletes’ attention was focused on their surroundings rather than being inhibited. Therefore, an athlete’s level of attention may increase while their level of alertness may decrease due to self-neglect. Generally, increased attention is associated with decreased alpha band power [24], while increased alertness is associated with increased beta band power [25]. Based on this knowledge, our first hypothesis was that alpha and beta band power decrease with the duration of endurance exercise, which differs from most sports and mental fatigue. The complexity of a biological system is indicative of its capacity to adapt and function in a changing environment [21]. Some diseases, such as Alzheimer’s disease, are known to result in a loss of complexity [23]. Therefore, our second hypothesis is that acute endurance exercise leads to changes in EEG complexity, which would be reflected in nonlinear eigenvalues. Finally, we will select eigenvalues that exhibit good linearity and sensitivity to further investigate the possibility of estimating athletes’ ability and performance.

## 2. Methods

### 2.1. Subjects

Subjects were recruited through invitations to participate in the study in a non-random manner. Subjects were required to meet the following inclusion criteria: endurance athletes over the age of 18 or prospective athletes over the age of 16 with at least two years of professional training, and no reported history of neurological or psychiatric illness or other pathological conditions that could potentially influence the outcomes of the study. A total of 24 participants who met the inclusion criteria were included in the study. One participant was excluded from the analysis due to the presence of incomplete data. The study was approved by the medical ethics committee of Jilin University, China, and was performed in accordance with relevant guidelines and regulations of the institutional review board after each subject had given written informed consent, and all procedures performed in studies involving human participants were in accordance with the Declaration of Helsinki.

### 2.2. Experimental Design and Procedure

A twenty-four-channel EEG measurement system (MBT-Smarting, Yu Shang Technology, Beijing, China) was used to acquire EEG signals from twenty-three subjects, induced to exhaustion by a power level progressive cycling test on a cycle ergometer (Ergoline Ergo Select 100 P, Bitz, Germany). The twenty-four channels were placed according to the international standard of 10–20. All of the subjects were asked to wash their hair before the EEG measurements to avoid extensive sweat or grease layers influencing the measurement results. In addition, the subjects breathed through a nonbreathable valve with the volume and the temperature of the expired air registered, so that the oxygen uptakes of subjects were calculated. The subjects were exhausted when they reached their maximal oxygen uptakes or their oxygen uptakes stopped increasing.

The subjects were fitted with EEG caps and breathing masks after a full warm-up. While hearing the starting instruction of the experimenter, the subjects pedaled the aerobic power cycling at a fixed power of 20 W for 1 min, in order to motivate the body and overcome physiological inertia, so as to achieve the best test state. Then, the initial power of the power car was set as 75 W for males and 50 W for females, with an increase of 25 W per 1.5 min for all subjects. The athletes were asked to ride until exhaustion with the rotation speed being kept within the range of 70~80 r per minute. The criteria for determining exhaustion include: (1) that the subject subjectively thought he was exhausted, (2) that the subject failed to maintain a 70~80 r/min motor rhythm, and (3) maximal oxygen uptake. After the subjects were exhausted, their EEG signals were measured for a five-minute finishing movement, and the power was set back to 20 W. The test procedure was shown in Figure 1.

The ‘.xdf’ file output by the EEG acquisition card was read. The 24-channel EEG signal was intercepted according to the rising edge marker signal, and the following EEG data were saved: the data for the 1-min low-power exercise, the data for the increasing power test, and the data for the 5 min finishing movement. The EEG acquisition frequency was 500 Hz.

### 2.3. Data Analysis

We wrote programs to process and analyze all the data with MATLAB 2021b (MathWorks, Natick, MA, USA).

#### 2.3.1. EEG Pre-Processing

Butterworth bandpass filters with cutoff frequencies of 1 to 80 Hz and notch filters of 50 Hz were used to filter EEG data. Independent component analysis (ICA) was used for the decomposition of EEG signals. A window with a length of 40 s and a step size of 39 s was employed to reduce the computational complexity of ICA decomposition process. A trained and experienced EEG expert was invited to identify motor artefacts, such as eye movements, contained in the signal components. Highly contaminated components were filtered out. Components suspected to be contaminated by motion artefacts or not highly contaminated were filtered using the multi-channel Wiener filtering method. The processed components were co-composed with other components to form the EEG signal. Data with bad channels were not considered in this study.

#### 2.3.2. Time-Frequency Eigenvalues

A sliding window with a length of 1000 (2 s) and a step size of 500 (1 s) was used to segment the data. Set matrix $T_{i}$ stands for the sliding windows’ center time matrix of subject  i, so $T_{i}=\left[\begin{array}{ccc} 1 & 2 & \begin{array}{cc} \cdots & N_{i} \end{array} \end{array}\right]$, where $N_{i}$ is the number of sliding windows.

Time-frequency eigenvalues for each segment of data were calculated, including MNF (Mean frequency), MDF (Median frequency), IMNF (Instantaneous mean frequency), IMDF (Instantaneous median frequency), and IAPF (Individual alpha peak frequency). Their generic formulas are presented in Appendix A for reference.

The MNF MDF, IMNF, IMDF, and IAPF matrix of subject i were set as ${MNF}_{i}$ ${MDF}_{i}$, ${IMNF}_{i}$, ${IMDF}_{i}$ and ${IAPF}_{i}$.

#### 2.3.3. Power Eigenvalues

Power eigenvalues include delta rhythm power, theta rhythm power, alpha rhythm power, beta rhythm power, and relative power of delta, theta, alpha, and beta rhythm. (Relative power of a rhythm is defined as a ratio between the power of this rhythm and the power of the total band, which ranges from 0.25 to 30 Hz).

The delta rhythm power is the power of EEG in the frequency band 0.25~4 Hz. The theta rhythm power is the power of EEG in the frequency band 4~8 Hz. The alpha rhythm power is the power of EEG in the frequency band 8~13 Hz. The delta rhythm power is the power of EEG in the frequency band 13~30 Hz.

We used $\delta_{i}$ for the delta rhythm power of subject i, which is the power of the data in the 0.25~4 Hz frequency band. Same things with $\theta_{i}$, $\alpha_{i}$, and $\beta_{i}$. $\delta /\left(\delta +\theta +\alpha +\beta \right)$ is the relative power of delta, and we used ${\delta R}_{i}$. Same things with ${\theta R}_{i}$, ${\alpha R}_{i}$, and ${\beta R}_{i}$.

#### 2.3.4. Nonlinear Eigenvalues

Nonlinear eigenvalues include Shannon entropy, Renyi entropy, Log energy entropy, time-varying log root, and Teager mean energy. Their generic formulas are presented in Appendix A for reference.

The Shannon entropy [26], Renyi entropy [27], log energy entropy [28], time-varying log root [29] and Teager energy [30] matrix of subject i were set as ${ShEn}_{i}$, ${ReEn}_{i}$, ${LogEn}_{i}$, ${LRSSV}_{i}$ and ${MTEn}_{i}$.

#### 2.3.5. Statistical Analysis

The EEG data included the data in motor phase and five-minute recovery phase. Correspondingly; the matrix $T_{i}$ was divided into $T_{i}^{seg1}=\left[\begin{array}{ccc} 1 & 2 & \begin{array}{cc} \cdots & N_{i}-300 \end{array} \end{array}\right]$ and $T_{i}^{seg2}=\left[\begin{array}{ccc} N_{i}-299 & N_{i}-298 & \begin{array}{cc} \cdots & N_{i} \end{array} \end{array}\right]$; where $N_{i}$ is the number of sliding windows. The eigenvalue matrices such as ${ShEn}_{i}$ are divided into ${ShEn}_{i}^{seg1}$ and ${ShEn}_{i}^{seg2}$ and so on

The data that have been divided are firstly tested for normality and homogeneity of variance. Box-Cox transform will be used to Normalize the data if they are not consistent with normal distribution and homogeneity of variance [31]. For variables that conformed to the normal distribution or just have been normalized, linear regression is used to analyze the correlation. The statistical test is bilateral, and the difference is statistically significant when the *p* value is less than 0.05.

The ratio ${post}^{-1\min}$ was defined as the ratio between the mean eigenvalues at region F in the last 1 min of motor phase and the mean eigenvalues in the first 1 min. The ratio ${post}^{+5\min}$ was defined as the ratio between the mean eigenvalues at zone F in the last 1 min of recovery phase and the mean eigenvalues in the first 1 min of motor phase.

The average regression coefficient of 24 channels was calculated with the correlation coefficient as the weight for EEG eigenvalues of each channel. Shannon entropy is taken as an example, and the method is shown in Formula (1). $\bar{{kShEn}_{i}^{seg1}}$ represents the mean regression coefficient of 24 channels in the motor phase of subject i. ${kShEn}_{\left(i,chnl\right)}^{seg1}$ represents the regression coefficient of channel chnl and subject i in the motor phase. ${RShEn}_{\left(i,chnl\right)}^{seg1}$ represents the correlation coefficient of channel chnl and subject i in the motor phase.
(1)$$\bar{{kShEn}_{i}^{seg1}}=\sum_{chnl=1}^{24}{kShEn}_{\left(i,chnl\right)}^{seg1}\times \frac{{RShEn}_{\left(i,chnl\right)}^{seg1}}{\sum_{j=1}^{24}{RShEn}_{\left(i,j\right)}^{seg1}}$$

The mean regression coefficients of EEG eigenvalues at zone F, T, P, O, and C of the brain were calculated by the same method, which were recorded as $\bar{{kShEn}_{i}^{seg1}\left(F\right)}$, $\bar{{kShEn}_{i}^{seg1}\left(T\right)}$ et al.

## 3. Results

### 3.1. Sample Features

Twenty-three cross-country skiing athletes (age, 18.00 ± 2.10 years; height, 174.62 ± 7.84 cm; weight, 64.63 ± 7.61 kg; body mass indexes, 21.20 ± 2.10 kg/m^2^) with no previous history of lower extremity or severe musculoskeletal injury participated in the study. Their basic information is shown in Table 1.

### 3.2. Test Performance

The duration of motor phase (excluding 5 min finishing movement) was 713.54 ± 187.84 s for male and 671.33 ± 122.26 s for female. According to the duration of exercise phase, the subjects were divided into Top group, Middle group, and Bottom group. The duration of motor phase of the three subjects’ groups was shown in Table 2.

### 3.3. Normalized EEG Eigenvalues

Table 3 presents the 18 EEG eigenvalues for the beginning and end of the motor phase and the end of the recovery phase in brain region F. Figure 2 shows the ratio ${post}^{-1min}$ and ${post}^{5min}$ for EEG eigenvalues in the F region.

### 3.4. Regression Analysis

Table 4 presents the regression coefficients of the EEG eigenvalues for male and female subjects during the motor and recovery phases. Figure 3 shows the regression and correlation coefficients for EEG eigenvalues of the 23 subjects during the motor phase and the recovery phase. The correlation coefficients of the nonlinear eigenvalues are higher than those of the power eigenvalues and time-frequency spectrum eigenvalues in both motion and recovery phases. Among all nonlinear eigenvalues, only ShEn and LogEn do not overlap in the distribution interval of regression coefficients in the two phases. Given the similarity of these two calculations, we chose the ShEn for additional study, which is more predictable because it falls during the motor phase and rises during the recovery phase.

### 3.5. Shannon Entropy

Figure 4 shows the ShEn trends in the F4 brain region for all subjects of different genders and performance groups. Figure 5 shows the regression coefficient for ShEn of the three groups subjects during the motor phase. The ShEn of each individual showed a clear trend, which was opposite in the exercise and recovery phase.

## 4. Discussion

The linearity and sensitivity of EEG eigenvalues were compared for the ride-to-exhaustion process, with the objective of identifying the most suitable eigenvalue for assessing the degree of exhaustion. The results indicated that the Shannon entropy is the most appropriate for this purpose. Further subgroup analyses of Shannon entropy revealed its potential to estimate athletic capacity.

### 4.1. Brain Study for Endurance Exercise

At the motor control level, athletic performance is contingent upon the programming of movements during the final stages of preparation for action [13]. A multitude of subsequent studies employing disparate experimental designs have demonstrated that skilled athletes exhibited a diminished complexity of event-related cortical activity in comparison to novices [32,33]. These findings align with the “neural efficiency” hypothesis, which was propounded by Haier and derived from investigations into the relationship between the brain and intelligence [34]. The neural efficiency of skilled athletes was found to be higher than that of novices, resulting in their cortical activity showing lower complexity in completing the same tasks as novices [13]. While the hypothesis has drawbacks, proponents and skeptics alike agree that accurate action programming leads to expert-like athletic performance. However, this premise requires verification for endurance sports. For instance, external attention methods have been shown to make runners more economical than internal attention methods [12]. A runner who focuses on their own fitness or posture is more likely to become exhausted than a runner who focuses on external factors such as the environment.

Recent studies have indicated that the electroencephalogram (EEG) pattern of subjects engaged in endurance exercise differs from that observed during other sports. Following endurance exercises such as cycling, subjects exhibited a shift towards higher frequencies in their IAPF [10]. Conversely, hockey players did not alter their IAPF when engaged in ice hockey, a team-skill game [15]. One principal method for improving performance in endurance sports is to enhance the economy of exercise, which is to utilize the minimum energy expenditure for optimal performance. Improving the sports economy encompasses a range of approaches, including physiological and psychological methods, long-term accumulation of sports, and the implementation of effective psychological mechanisms. Acute or chronic exercise has been shown to have a positive impact on brain function, which is associated with improvements in cognitive ability [14,35,36,37]. This, in turn, affects athletic performance [38]. Therefore, it was hypothesized that the EEG eigenvalues of athletes of different levels would also present certain regular changes in the process of performing endurance sports tasks.

### 4.2. Eigenvalues Trends

In this study, twenty-three cross-country skiers or would-be athletes completed an endurance cycling task with increasing power levels. According to the self-reports, motor performance, and respiratory entropy of subjects, it is evident that every subject was exhausted. The subjects were divided into three groups according to their performance in this test: Top group, Middle group, and Bottom group. The Top group is comprised of elite and Class I athletes, the Middle group is comprised of Class II athletes, and the Bottom group is comprised of would-be athletes. The EEG of subjects was recorded throughout the test. Three EEG eigenvalues were calculated: time-frequency eigenvalues, power eigenvalues and nonlinear eigenvalues. The sensitivity and linearity of EEG eigenvalues to endurance exercise duration are two important criteria for considering the suitability of estimating endurance exercise capacity. The sensitivity was characterized using the magnitude of the regression coefficients difference between the motor and recovery phases and between the F and other regions of the brain. The linearity was characterized by the correlation coefficient between the motor and recovery phases.

With regard to the results, it was observed that the time-frequency eigenvalues, except for the IAPF, exhibited no significant differences. Their variations were observed only in certain subjects. The IAPF demonstrated a gradual shift toward higher frequency, indicative of the test being an exhaustive exercise for the subjects. However, this upward trend in the IAPF was observed to be slow and concentrated primarily during the end of the motor phase and in the recovery phase. This suggests that the IAPF is hypersensitive to the duration of endurance exercise.

The relative power of all bands except the delta rhythm demonstrated statistically significant differences between the two time points in the power eigenvalues method. The relative power of theta, alpha, and beta rhythms decreased during the motor phase and recovered to a value near or above the initial value during the recovery phase. Previous findings and hypotheses indicate that the decrease in alpha power may be indicative of an increase in the number of activated cortical neurons with exercise duration [24,39,40]. The decrease in theta energy suggests that the subjects exhibited no tendency to fall asleep [8]. A reduction in beta energy indicates a decline in arousal levels within the brain [25]. Consequently, the subjects’ attention was observed to increase, despite a reduction in arousal, which could be attributed to participants directing their attention elsewhere during cycling tasks, as elite athletes tend to do during endurance exercise. However, the change in power eigenvalues was lagging in both the motor and recovery phases, indicating that the linearity was low.

In both the motor and recovery phases, the nonlinear eigenvalues exhibited higher regression and correlation coefficients than the power and time-frequency eigenvalues. Among all the nonlinear eigenvalues, only ShEn and LogEn exhibited completely opposite regression coefficients in both phases, indicating that their trends were entirely distinct in both phases. ShEn was chosen for further investigation because it exhibits a more predictable pattern than LogEn. This is due to the fact that ShEn undergoes a decline during the motor phase and an increase during the recovery phase, while the opposite is true for LogEn.

In the motor phase, the ShEn for the Top group demonstrated a rate of decrease similar to that of the Middle group, which was notably slower than that of the Bottom group. The fastest rate of decline in ShEn for the Bottom group occurred at the shortest exercise duration. The lower endpoints of the decline in ShEn for the Top group occurred at the longest exercise duration compared to the Middle group. In addition, the standard deviation of the 24-channel weighted mean regression coefficient for subjects in the Bottom group was larger than that for the other two groups. There was a notable difference between the weighted mean regression coefficient for the F region and the other four regions. The average value for the entire region was comparable to the average value for the F region, yet it was significantly different from the average value for the other regions. These phenomena were attributed to the substantial discrepancy between the ShEn of the F region and the other regions of the brain in subjects in the Bottom group, with the correlation coefficient of the F region being higher than that of the other regions. The F region is responsible for motor function in the brain. Therefore, it is reasonable to observe differences between the F region and other regions during endurance cycling. These differences are primarily concentrated in the Bottom group of subjects.

The impact of psychological factors on motor performance during repetitive work represents a boundary case in the field of brain–behavior connections. The quantification of this effect by EEG eigenvalue calculations in this study contributes to the understanding of the brain–behavior connections. Furthermore, this study contributes to the reduction of sport injury risk. The continuous accumulation of fatigue due to overloaded training plans under the pressure of competitive performance is a major cause of injury in athletes. The estimation of performance is thought to be helpful in predicting fatigue levels as well as balancing training and competition schedules.

However, there are several limitations in this study. In the second half of the study, we divided the subjects into three groups of each gender based on their motor performance. According to sample estimation criteria, we need at least 36 subjects to yield a statistically significant test. In future studies, the sample size should be increased in order to preferably reduce inter-subject differences. Consequently, further investigation of Shannon entropy is required, as the physical meaning of its trends remains unclear. In future studies, we intend to examine the brain–behavior relationship in individuals who do not excel in endurance sports.

## 5. Conclusions

In this study, we investigated the trends of 18 EEG eigenvalues during an exhaustive endurance exercise session for the purpose of endurance exercise performance versus potential research. The results showed that, unlike precision sports such as shooting and acrobatics, subjects’ arousal levels decreased and attention levels increased with increasing exercise duration.

In addition, the sensitivity and linearity of EEG nonlinear eigenvalues affected by endurance exercise duration were higher than those of time-frequency eigenvalues and power eigenvalues. Among them, Shannon entropy is the most suitable eigenvalue to be applied to guide endurance exercise training. It has an opposite trend in the exercise phase to the recovery phase, with a stronger trend in the F region of the brain than in other regions. Moreover, Shannon’s entropy has room for mathematical improvement, and it is of great interest to use variations of Shannon’s entropy for research.

## Figures and Tables

**Figure 1 brainsci-14-01135-f001:**
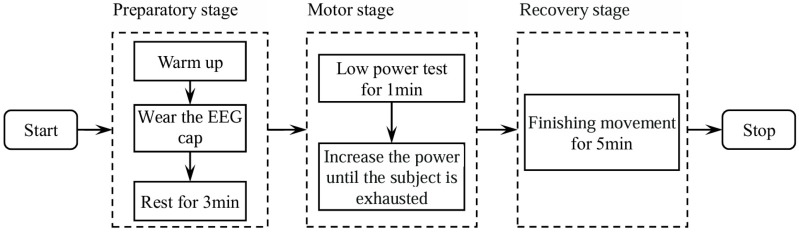
Flow chart for the cycling test.

**Figure 2 brainsci-14-01135-f002:**
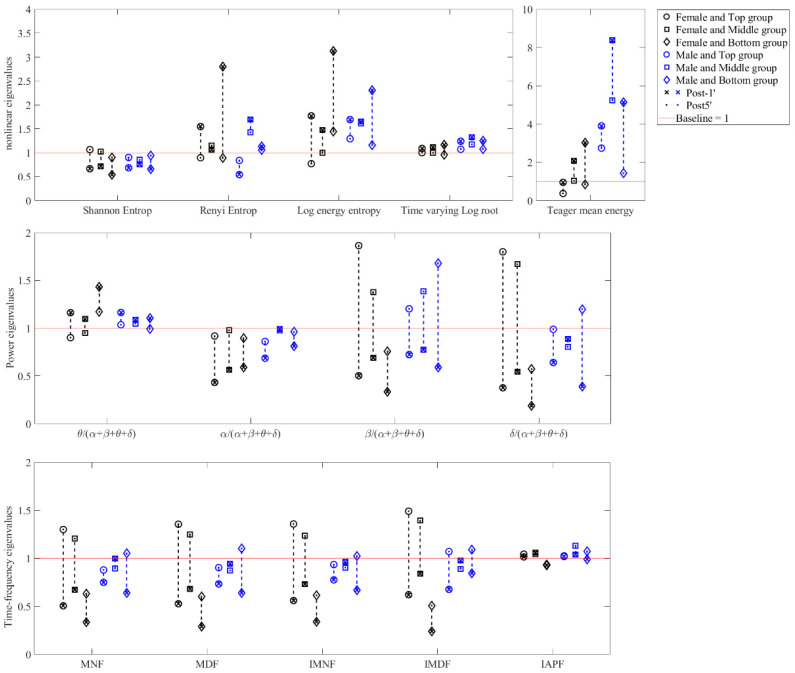
The ratio ${post}^{-1min}$
and ${post}^{5min}$ for EEG eigenvalues in the F region of the brain.

**Figure 3 brainsci-14-01135-f003:**
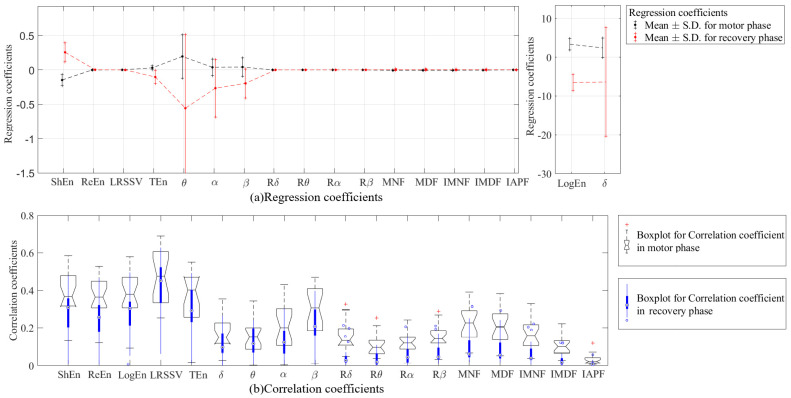
The regression and correlation coefficients for each EEG eigenvalues of the 23 subjects during the motor phase and the recovery phase. (**a**) Mean ± S.D. of regression coefficients. (**b**) Box−plot for correlation coefficients in motor and recovery phase.

**Figure 4 brainsci-14-01135-f004:**
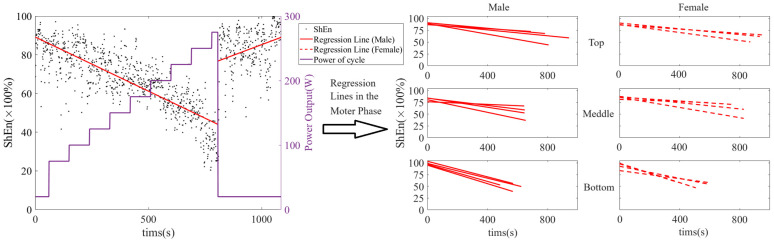
The Shannon entropy trends in the F4 brain region. (The Shannon entropy trends in the motor and recovery phases of a single subject are illustrated in the left panel. The black points represent Shannon entropy data, whereas the red lines indicate the regression lines for time and Shannon entropy during these phases. The purple line represents the bicycle power output. The right panel depicts the Shannon entropy trends for all subjects of different genders and performance groups).

**Figure 5 brainsci-14-01135-f005:**
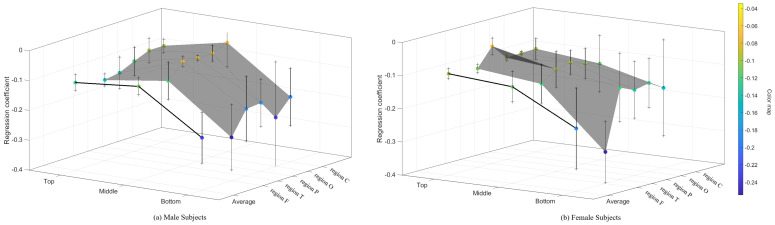
The regression coefficients for Shannon entropy of five brain regions of all three groups subjects during the motor phase.

**Table 1 brainsci-14-01135-t001:** Basic information of subjects.

Sample Size	Age (Year)	Height (cm)	Weight (kg)	Time of Training (Year)
Male (*n* = 11)	17.1 ± 2.3	179.4 ± 7.5	68.0 ± 7.5	2.0 ± 1.3
Female (*n* = 12)	18.8 ± 1.5	169.8 ± 4.7	61.3 ± 6.4	3.4 ± 1.2

**Table 2 brainsci-14-01135-t002:** Athletic performance of subjects in different groups.

	Male Subjects			Female Subject		
	Top Group	Middle Group	Bottom Group	Top Group	Middle Group	Bottom Group
N	3	4	4	4	4	4
Athletic performance(s)	922.00 ± 47.47	803.00 ± 40.71	534.80 ± 87.88	843.33 ± 86.29	650.67 ± 20.93	540.67 ± 49.09

**Table 3 brainsci-14-01135-t003:** The EEG eigenvalues during the first and last minutes of the motor phase and the last minute of the recovery phase in brain region F.

Eigenvalues (Unit)	Female			Male		
	+0 min	P-6 min	P-1 min	+0 min	P-6 min	P-1 min
ShEn (×1)	253.94 ± 141.19	167.12 ± 102.91	246.49 ± 125.83	262.84 ± 95.20	182.26 ± 74.33	236.82 ± 86.92
ReEn (×1)	1.21 ± 1.11	1.13 ± 0.76	1.15 ± 1.05	1.17 ± 0.79	0.85 ± 0.62	1.07 ± 0.72
LogEn (×10^4^)	2.14 ± 1.08	3.93 ± 1.87	2.14 ± 1.05	1.83 ± 0.79	3.18 ± 0.90	2.32 ± 0.92
LRSSV (×1)	1.33 ± 0.16	1.49 ± 0.14	1.31 ± 0.11	1.171 ± 0.20	1.47 ± 0.21	1.27 ± 0.17
MTEn (×1)	2.30 ± 3.05	2.49 ± 1.33	1.01 ± 0.45	1.14 ± 2.09	3.73 ± 3.60	1.89 ± 2.47
δ (${\mu V}^{2}$)	99.73 ± 128.66	441.50 ± 496.67	66.03 ± 66.28	36.35 ± 37.78	263.90 ± 324.22	152.65 ± 238.50
θ (${\mu V}^{2}$)	34.30 ± 82.73	22.53 ± 14.76	7.99 ± 6.69	6.01 ± 7.39	31.77 ± 41.93	21.55 ± 36.24
α (${\mu V}^{2}$)	17.93 ± 49.66	7.17 ± 3.62	2.44 ± 0.92	2.90 ± 5.45	14.41 ± 19.52	7.75 ± 11.96
β (${\mu V}^{2}$)	4.69 ± 7.92	4.62 ± 2.27	2.04 ± 0.78	2.85 ± 6.27	8.04 ± 8.44	4.18 ± 6.05
δR (%)	73.66 ± 10.14	88.24 ± 5.74	72.77 ± 11.79	75.26 ± 10.16	83.58 ± 7.39	76.25 ± 6.95
θR (%)	14.66 ± 02.48	7.95 ± 3.50	13.48 ± 3.06	14.99 ± 5.20	10.77 ± 2.46	12.73 ± 3.05
αR (%)	7.06 ± 3.90	3.41 ± 1.95	8.07 ± 4.21	7.43 ± 3.71	4.66 ± 2.62	8.91 ± 3.32
βR (%)	8.68 ± 6.15	2.71 ± 1.64	9.34 ± 6.61	6.69 ± 3.53	4.04 ± 3.74	5.92 ± 2.67
IMNF (Hz)	7.81 ± 3.39	3.72 ± 1.20	7.40 ± 2.77	6.37 ± 1.88	4.78 ± 2.03	5.74 ± 1.45
IMDF (Hz)	6.74 ± 3.43	3.04 ± 0.90	6.35 ± 2.84	5.34 ± 1.63	3.93 ± 1.81	4.90 ± 1.32
MNF (Hz)	6.51 ± 3.14	3.25 ± 0.91	6.22 ± 2.62	5.33 ± 1.67	4.07 ± 1.70	4.88 ± 1.23
MDF (Hz)	2.95 ± 2.04	1.31 ± 0.30	2.65 ± 1.63	2.24 ± 0.96	1.702 ± 0.57	2.16 ± 0.82
IAPF (Hz)	9.62 ± 0.74	9.64 ± 0.71	9.70 ± 0.62	9.21 ± 0.48	9.33 ± 0.50	9.85 ± 0.43

Note: ‘+0 min’: First minute of the motor phase; ‘P-6 min’: Last minute of the motor phase; ‘P-1 min’: Last minute of the recovery phase.

**Table 4 brainsci-14-01135-t004:** Regression coefficients of EEG eigenvalues in the motor and recovery phases.

Eigenvalues	Female		Male	
	Motor Phase	Recovery Phase	Motor Phase	Recovery Phase
ShEn (×1)	−0.14 ± 0.07	0.28 ± 0.15	−0.15 ± 0.09	0.24 ± 0.13
ReEn (×10^−4^)	−4.09 ± 6.80	8.16 ± 12.95	−7.33 ± 4.55	11.26 ± 5.43
LogEn (×1)	28.63 ± 12.31	−55.48 ± 16.15	28.08 ± 16.49	−56.60 ± 34.29
LRSSV (×10^−4^)	4.44 ± 1.43	−12.35 ± 2.36	5.57 ± 1.78	−12.69 ± 4.22
MTEn (×10^−2^)	3.88 ± 2.02	−11.70 ± 10.57	2.11 ± 3.24	−9.09 ± 9.61
δ (×1)	1.56 ± 1.01	−3.11 ± 2.70	3.16 ± 3.20	−9.34 ± 19.12
θ (×1)	0.12 ± 0.11	−0.31 ± 0.25	0.26 ± 0.42	−0.78 ± 1.46
α (×10^−2^)	4.86 ± 9.85	−22.91 ± 28.83	2.75 ± 14.50	−29.92 ± 52.16
β (×10^−2^)	7.84 ± 4.88	−22.04 ± 21.94	0.86 ± 17.63	−17.46 ± 21.26
δR (×10^−4^)	2.56 ± 2.17	−4.54 ± 3.72	1.39 ± 2.42	−2.56 ± 3.43
θR (×10^−4^)	−1.23 ± 0.39	1.14 ± 2.45	−0.52 ± 1.30	−0.08 ± 2.48
αR (×10^−5^)	−6.11 ± 11.18	13.47 ± 18.69	−4.74 ± 9.07	30.34 ± 20.50
βR (×10^−4^)	−1.05 ± 1.20	1.90 ± 2.28	−0.45 ± 0.84	−0.29 ± 2.76
IMNF (×10^−3^)	−7.75 ± 4.82	8.66 ± 10.25	−3.66 ± 4.70	−1.13 ± 13.06
IMDF (×10^−3^)	−7.53 ± 5.28	8.58 ± 10.76	−3.40 ± 4.53	−1.24 ± 12.69
MNF (×10^−3^)	−6.26 ± 5.07	6.96 ± 10.68	−2.63 ± 4.17	−2.74 ± 12.04
MDF (×10^−3^)	−5.19 ± 5.09	6.73 ± 11.00	−0.76 ± 2.46	−1.53 ± 8.30
IAPF (×10^−5^)	4.38 ± 14.91	−15.87 ± 46.75	3.33 ± 16.78	32.64 ± 20.92

## Data Availability

The datasets used and/or analyzed during the current study are available from the corresponding author upon reasonable request.

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
