# Peer review of "The Electroencephalogram (EEG) Study for Estimating Endurance Sports Performance Base on Eigenvalues Extraction Method"

_brainsci, 2024, doi:10.3390/brainsci14111135_

Round 1
Reviewer 1 Report
Comments and Suggestions for Authors
see attached

Author Response
Comment 1:
"Introduce this acronym."
Response:
Thanks for your suggestion. We have included the full spelling of “EEG”.
Comment 2:
"Brief overview about endurance exercise is recommended here."
Response:
Thanks for your proposed modification. We have included the brief overview about endurance exercise in the position you suggested.
Comment 3:
"This term appeared suddenly."
Response:
Thanks for your reminder. We have inserted the abbreviation cross-reference table at the beginning of the article and made it clear in the text that IAPF is an eigenvalue of EEG.
Comment 4:
"List the inclusion and exclusion criteria."
Response:
Thanks for your proposed modification. We have listed the inclusion criteria for the subjects. Newly added text is displayed in blue.
" Subjects were required to meet the following inclusion criteria: endurance athletes over the age of 18 or prospective athletes over the age of 16 with at least two years of professional training, and no reported history of neurological or psychiatric illness or other pathological conditions that could potentially influence the outcomes of the study."
Comment 5:
"State the country."
Response:
Thanks for your suggestion. We have stated the country in which the test was taken.
Comment 6 & 8:
"Move these information to the results section."
"Summarize the key features of the study participants at the beginning of this section."
Response:
Thanks for your proposed modification. We have moved that information to the beginning of the results section.
Comment 7:
"I found this section to be overly lengthy due to an excess of details, which makes it difficult to follow. I would recommend summarizing the key stats/calculations used and moving the more specific details to the supplementary materials."
Response:
Thanks for your comments. We have summarized the key calculations and included the details in Appendix A. Newly added text is displayed in blue and moved text is displayed on a yellow background to make it easier for the reader to identify.
Comment 9:
"An opening paragraph containing a clear summary of the main findings is recommended "
Response:
Thanks for your suggestion. We have included a summary of the main findings between 4 and 4.1.
Comment 10:
"Authors are recommended to discuss the broader implications of the current study findings and future directions"
Response:
Thanks for your suggestion. We have included the main contributions of the findings to the field of neurological and motor training at the end of the discussion section. We have also included future directions based on the limitations of the article.
Reviewer 2 Report
Comments and Suggestions for Authors
A crucial question for this manuscript is the extent to which electrical artifacts have been properly corrected in the dataset. As motion and motor artifacts created electrical noise upwards of one magnitude larger than electrical sources from the brain, the reader should be confident that such electrical noise is not present in the data. For example, it’s not hard to imagine that as one is cycling for long periods of time, there might be changes in one’s motor activity that linearly change over time (e.g., becoming less efficient at pedaling; flailing more with one’s limbs while struggling to finish) which could reasonably lead to a change in electrical noise detected at the scalp. Similarly, eye movement could systematically change over the course of an endurance effort, which would create a huge electrical distortion dwarfing any neural signals under the scalp that might be changing.
Crucially, previous EEG work in this field has relied on resting measures of EEG, as EEG is notoriously confounded by electrical noise when participants are moving. Yet, the current manuscript analyzes EEG in participants engaged in a very high degree of motor function. What precedent is there for this? What steps have the authors taken to ensure that their data isn’t contaminated by motion artifact?
The answer is that authors pay zero attention to how artifacts are corrected in their EEG data. This is absolutely unprecedented in EEG research, even when participants are seated perfectly still. The fact that the authors pay no attention to artifact correction of EEG is a major warning sign that the EEG methodology here is extremely problematic. What other errors in standard EEG methodology may be present in the data (for which none of the raw EEG waveforms are available for the reader to inspect)? The math in the paper is fancy, but it is all meaningless if not being used on a clean EEG signal that we are confident is uncontaminated by motor artifact.
Absent of the authors sharing their raw data and custom Matlab scripts, showing that they are indeed artifact correcting their data and in a manner that seems to reasonably remove electrical artifact, this manuscript has no right to be published as reflecting human EEG. Perhaps one could do these math techniques on simulated data, but there is zero reason to believe the results coming out of these models exclusively reflect human brain activity.
I could be inclined to be more forgiving, however, the authors cite two previous studies they claim have measured EEG during endurance activity, as a basis for their using a “similar” methodology. Yet, neither of the two studies cited measured EEG during exercise, and one of these studies is of hockey players shooting pucks (not endurance activity). The authors claim that these two papers show different patterns of EEG for endurance activity, but neither study cited actually makes these claims.
Taken together with the above, this is a blatant regard for previous literature. Scientific progress is made incrementally, by adopting the methods of work that came before and refining it. Failing to adopt the most basic aspects of the experimental methodology (i.e., artifact correction) leads me to lose all trust in the scientific rigour of this manuscript.
Comments on the Quality of English Language
The quality of english is good.
Author Response
Comments to the Author:
A crucial question for this manuscript is the extent to which electrical artifacts have been properly corrected in the dataset. As motion and motor artifacts created electrical noise upwards of one magnitude larger than electrical sources from the brain, the reader should be confident that such electrical noise is not present in the data. For example, it’s not hard to imagine that as one is cycling for long periods of time, there might be changes in one’s motor activity that linearly change over time (e.g., becoming less efficient at pedaling; flailing more with one’s limbs while struggling to finish) which could reasonably lead to a change in electrical noise detected at the scalp. Similarly, eye movement could systematically change over the course of an endurance effort, which would create a huge electrical distortion dwarfing any neural signals under the scalp that might be changing.
Crucially, previous EEG work in this field has relied on resting measures of EEG, as EEG is notoriously confounded by electrical noise when participants are moving. Yet, the current manuscript analyzes EEG in participants engaged in a very high degree of motor function. What precedent is there for this? What steps have the authors taken to ensure that their data isn’t contaminated by motion artifact?
The answer is that authors pay zero attention to how artifacts are corrected in their EEG data. This is absolutely unprecedented in EEG research, even when participants are seated perfectly still. The fact that the authors pay no attention to artifact correction of EEG is a major warning sign that the EEG methodology here is extremely problematic. What other errors in standard EEG methodology may be present in the data (for which none of the raw EEG waveforms are available for the reader to inspect)? The math in the paper is fancy, but it is all meaningless if not being used on a clean EEG signal that we are confident is uncontaminated by motor artifact.
Absent of the authors sharing their raw data and custom Matlab scripts, showing that they are indeed artifact correcting their data and in a manner that seems to reasonably remove electrical artifact, this manuscript has no right to be published as reflecting human EEG. Perhaps one could do these math techniques on simulated data, but there is zero reason to believe the results coming out of these models exclusively reflect human brain activity.
I could be inclined to be more forgiving, however, the authors cite two previous studies they claim have measured EEG during endurance activity, as a basis for their using a “similar” methodology. Yet, neither of the two studies cited measured EEG during exercise, and one of these studies is of hockey players shooting pucks (not endurance activity). The authors claim that these two papers show different patterns of EEG for endurance activity, but neither study cited actually makes these claims.
Taken together with the above, this is a blatant regard for previous literature. Scientific progress is made incrementally, by adopting the methods of work that came before and refining it. Failing to adopt the most basic aspects of the experimental methodology (i.e., artifact correction) leads me to lose all trust in the scientific rigour of this manuscript.
Responses to the comments:
Thank you for pointing out the lack of a method for filtering out motion artifacts in the article. In previous studies, EEG data has been predominantly acquired during periods of rest, such as while observing motion [1], viewing images [2], or during passive recovery [3]. This is sufficient to explore the effects of acute exhaustion exercise on the brain. The benefits of acute endurance exercise for both the body and the brain are well documented [4,5] and justify its promotion to the general public. However, it is important to acknowledge the inherent risk of sports injuries, even for those under the guidance of professional athletes[6,7]. In order to reduce injury risk during exercise, it would be beneficial to have an intuitive method of assessing the level of exertion, which could then be used to guide athletes or the general public in their training. It is clear that the acquisition of EEG data before and after exercise alone will not be sufficient for this purpose.
We know that the acquisition of EEG signals during motion can be confounded by various types of noise. So, we reviewed the research methods and signal processing in previous studies of EEG during movement. And based on the results of the review, we did a lot of work to reduce and filter the motor artifacts. We would like to address our work on reducing motion artifacts in 1) the motion task and 2) data processing. In addition, as you suggested, we have uploaded the minimal raw data set and the programs.
1) The motion task
Firstly, our choice of a cycling test to induce force exhaustion in subjects was well considered. Many references have been consulted for the measurement of EEG signals during cycling [8-12]. One [12] of the articles used a similar experimental paradigm. "EEG recordings were acquired during each phase. The five phases were characterized by the following conditions and tasks: Phase I (baseline condition), Phase II (pre-cycling condition), Phase III (cycling condition), Phase IV (active recovery condition), and Phase V (passive recovery condition)."
In addition, the cycling task is an experimental paradigm with many of the strict constraints commonly used in exercise research. Even in tests without EEG recording, the buttocks are not allowed to be out of the saddle and the torso is also not allowed to sway significantly.
Secondly, we used endless riding tasks. Some experienced trainers will use the inability to complete an athletic task despite the intense demands of others as a criterion for exhaustion. Some athletes would do the same, even though we have forbidden subjects from using their voices to motivate themselves. Endless riding tasks can make this ineffective or even counterproductive, and elite athletes know this. Endless riding tasks therefore help to test the calmness of the whole field.
Thirdly, we used a breathing mask to objectify the exhaustion standard, which would also suggest at a mental level that subjects should reduce the amplitude of head shaking.
2) Data processing.
Even when tested in strict accordance with experimental paradigms, data contamination of EEG cannot be ignored. Therefore, filtering of EEG signals is essential. Common filtering methods include WT [13,14], ICA [15,16], EMD [11], and their extensions. These methods are characterized by the decomposition of the signal based on mathematical features. Since artifacts such as EMG, ECG, and ocular artifacts have their sources, they are suitable to be filtered out by signal decomposition methods. However, the problem is that these artifacts are difficult to detect adaptively and completely through intuitive features and require visual recognition by trained experts.
In this study, the ICA was used for the decomposition of EEG signals. A window with a length of 40s and a step size of 39s was employed to reduce the computational complexity of ICA decomposition process. A trained and experienced EEG expert was invited to identify motor artifacts visually in the signal components. Highly contaminated components were filtered out. Components suspected to be contaminated by motion artifacts or not highly contaminated were filtered using the multi-channel Wiener filtering method. The multi-channel Wiener filter is commonly used in geological exploration data processing and is suitable for filtering out noise from the same source in multi-channel signals. The processed components were co-composed with other components to form the EEG signal. Data with bad channels were not considered in this study.
Although we completed the filtering of motion artefacts, I did not include it in the manuscript due to my neglect of the purpose of this work. In this revision, we have included the above work in the manuscript and shown the added text in blue.
In summary, we have done a lot of effective work to reduce motion artifacts.
We thank you again for your rigorous and constructive suggestions. Your suggestions not only make our articles more rigorous and better, but also serve as a reminder for our future research. It was an honor for us to have your advice. If you deem it necessary to add other revisions to the manuscript, we would be willing to carry out additional work to improve the results.
*******************************
References
- Kaneko, N.; Yokoyama, H.; Masugi, Y.; Watanabe, K.; Nakazawa, K. Phase dependent modulation of cortical activity during action observation and motor imagery of walking: An EEG study. Neuroimage 2021, 225, doi:10.1016/j.neuroimage.2020.117486.
- Cona, G.; Cavazzana, A.; Paoli, A.; Marcolin, G.; Grainer, A.; Bisiacchi, P.S. It's a Matter of Mind! Cognitive Functioning Predicts the Athletic Performance in Ultra-Marathon Runners. Plos One 2015, 10, doi:10.1371/journal.pone.0132943.
- Gutmann, B.; Mierau, A.; Huelsduenker, T.; Hildebrand, C.; Przyklenk, A.; Hollmann, W.; Strueder, H.K. Effects of Physical Exercise on Individual Resting State EEG Alpha Peak Frequency. Neural Plasticity 2015, 2015, doi:10.1155/2015/717312.
- Parry-Williams, G.; Sharma, S. The effects of endurance exercise on the heart: panacea or poison? Nature Reviews Cardiology 2020, 17, 402-412, doi:10.1038/s41569-020-0354-3.
- Hillman, C.H.; Erickson, K.I.; Kramer, A.F. Be smart, exercise your heart: exercise effects on brain and cognition. Nature Reviews Neuroscience 2008, 9, 58-65, doi:10.1038/nrn2298.
- Almekinders, L.C.; Engle, C.R. Common and Uncommon Injuries in Ultra-endurance Sports. Sports Medicine and Arthroscopy Review 2019, 27, 25-30, doi:10.1097/jsa.0000000000000217.
- Egger, A.C.; Oberle, L.M.; Saluan, P. The Effects of Endurance Sports on Children and Youth. Sports Medicine and Arthroscopy Review 2019, 27, 35-39, doi:10.1097/jsa.0000000000000230.
- Pontifex, M.B.; Hillman, C.H. Neuroelectric and behavioral indices of interference control during acute cycling. Clinical Neurophysiology 2007, 118, 570-580, doi:10.1016/j.clinph.2006.09.029.
- Chang, Y.-K.; Pesce, C.; Chiang, Y.-T.; Kuo, C.-Y.; Fong, D.-Y. Antecedent acute cycling exercise affects attention control: an ERP study using attention network test. Frontiers in Human Neuroscience 2015, 9, doi:10.3389/fnhum.2015.00156.
- Olson, R.L.; Chang, Y.-K.; Brush, C.J.; Kwok, A.N.; Gordon, V.X.; Alderman, B.L. Neurophysiological and behavioral correlates of cognitive control during low and moderate intensity exercise. Neuroimage 2016, 131, 171-180, doi:10.1016/j.neuroimage.2015.10.011.
- Zink, R.; Hunyadi, B.; Van Huffel, S.; De Vos, M. Mobile EEG on the bike: disentangling attentional and physical contributions to auditory attention tasks. Journal of Neural Engineering 2016, 13, doi:10.1088/1741-2560/13/4/046017.
- di Fronso, S.; Fiedler, P.; Tamburro, G.; Haueisen, J.; Bertollo, M.; Comani, S. Dry EEG in Sports Sciences: A Fast and Reliable Tool to Assess Individual Alpha Peak Frequency Changes Induced by Physical Effort. Frontiers in Neuroscience 2019, 13, doi:10.3389/fnins.2019.00982.
- Al-Qazzaz, N.K.; Ali, S.H.B.M.; Ahmad, S.A.; Islam, M.S.; Escudero, J. Automatic Artifact Removal in EEG of Normal and Demented Individuals Using ICA-WT during Working Memory Tasks. Sensors 2017, 17, doi:10.3390/s17061326.
- Alyasseri, Z.A.A.; Khader, A.T.; Al-Betar, M.A.; Acm. Optimal Electroencephalogram Signals Denoising using Hybrid β-Hill Climbing Algorithm and Wavelet Transform. In Proceedings of the International Conference on Imaging, Signal Processing and Communication (ICISPC), Penang, MALAYSIA, 2015
Jul 26-28, 2017; pp. 106-112.
- Pontifex, M.B.; Gwizdala, K.L.; Parks, A.C.; Billinger, M.; Brunner, C. Variability of ICA decomposition may impact EEG signals when used to remove eyeblink artifacts. Psychophysiology 2017, 54, 386-398, doi:10.1111/psyp.12804.
- Li, Y.; Wang, P.T.; Vaidya, M.P.; Flint, R.D.; Liu, C.Y.; Slutzky, M.W.; Do, A.H. Electromyogram (EMG) Removal by Adding Sources of EMG (ERASE)-A Novel ICA-Based Algorithm for Removing Myoelectric Artifacts From EEG. Frontiers in Neuroscience 2021, 14, doi:10.3389/fnins.2020.597941.
Reviewer 3 Report
Comments and Suggestions for Authors
The authors present an EEG Study to estimate performance in endurance sports based on the eigenvalue extraction method. In principle the topic could be of interest to readers, however there are details that need to be improved.
EGG should not appear in the title and abstract without explaining what it refers to.
The summary provides clear information and the key words have been well chosen.
The introduction is clear and well structured.
Regarding the selection of participants, if it was not random, how were they contacted? Did you contact that group of participants for any specific reason?
The results are well described and clear to understand.
The discussion is clear, and well done, but it should incorporate some limitations.
The conclusions are very long, summarize it a little more
Author Response
Comments to the Author:
The authors present an EEG Study to estimate performance in endurance sports based on the eigenvalue extraction method. In principle the topic could be of interest to readers, however there are details that need to be improved.
Responses to the comments:
Thank you very much for your detailed review and specific comments on our manuscript. We have duly followed your comments and suggestions to address your concerns. We hope you find our revision satisfactory.
Reviewer's comments and responses:
Comment 1:
"EGG should not appear in the title and abstract without explaining what it refers to."
Response:
Thanks for your suggestion. We've made a few changes to the title, as you can see below. The newly added text is displayed in blue.
Title: The Electroencephalogram (EEG) Study for Estimating Endurance Sports Performance Base on Eigenvalues Extraction Method
Comment 2 & 3:
"The summary provides clear information and the key words have been well chosen."
"The introduction is clear and well structured."
Response:
Thank you so much for the recognition of our article!
Comment 4:
"Regarding the selection of participants, if it was not random, how were they contacted? Did you contact that group of participants for any specific reason?"
Response:
Thanks for your reminder. We obtained contact details of potential subjects by contacting national sports teams and club directors. Taking into account the familiarity with and implementation of endurance exercise strategies, we used age and years of training as inclusion criteria. Subjects were required to meet the following inclusion criteria: endurance athletes over the age of 18 or prospective athletes over the age of 16 with at least two years of professional training, and no reported history of neurological or psychiatric illness or other pathological conditions that could potentially influence the outcomes of the study. Priority will be accorded to those who have undergone closed training and whose future career plans were not in conflict with the test.
The above has been simplified and included in the article.
Comment 5:
"The results are well described and clear to understand."
Response:
We appreciate your approval!
Comment 6:
"The discussion is clear, and well done, but it should incorporate some limitations."
Response:
Thanks for your proposed modification. We have added article limitations at the end of the discussion section.
Comment 7:
"The conclusions are very long, summarize it a little more"
Response:
Thanks for your reminder. We have simplified the conclusion section.